# Whole-Genome Bisulfite Sequencing Reveals a Role for DNA Methylation in Variants from Callus Culture of Pineapple (*Ananas comosus* L.)

**DOI:** 10.3390/genes10110877

**Published:** 2019-11-01

**Authors:** Wenqiu Lin, Xi’ou Xiao, Hongna Zhang, Yunhe Li, Shenghui Liu, Weisheng Sun, Xiumei Zhang, Qingsong Wu

**Affiliations:** 1South Subtropical Crop Research Institute, Chinese Academy of Tropical Agricultural Sciences, Zhanjiang 524091, China; linwenqiu1989@163.com (W.L.); xiao-forlearning@163.com (X.X.); 13692476979@139.com (H.Z.); Liyunhe16@163.com (Y.L.); Lizx200@126.com (S.L.); sunweisheng1234@sina.com (W.S.); asiazhang1975@163.com (X.Z.); 2Laboratory of Tropical Fruit Biology, Ministry of Agriculture, Zhanjiang 524091, China

**Keywords:** pineapple, DNA methylation, somaclonal variation, bisulfite sequencing, callus culture

## Abstract

DNA methylation changes can occur in some loci during callus culture, resulting in somaclonal variations (SVs). In the present study, we applied whole genome bisulfite sequencing to analyze context-specific DNA methylation changes in the pineapple genome between the cutting seedings and 5 SV plants. In general, SV plants exhibited methylation patterns analogous to those of cutting seedlings (CK). A total of 27.98% of the genomic cytosines of CK were methylcytosines, which was higher than that of 5 SV plants. Moreover, mCG and mCHG was hypermethylated, whereas mCHH was hypomethylated among the 5 SV plants genomic when compared with the CK. Most of the variation of DNA methylation was distributed in gene bodies, thus suggesting that phenotypic differences are probably perturbed by genes methylated from callus culture. In addition, the methylated genes were highly enriched for the Gene Ontology (GO) categories of binding and catalytic activity, cell part and organelle, cellular process, abiotic stimulus, and DNA modification. These results suggest that methylation mediates these pathways in the callus culture of pineapple. The results also suggested that the callus culture induced DNA methylation may result in the SV.

## 1. Introduction

DNA methylation is an epigenetic modification that plays a critical role in diverse biological processes, including regulation of gene expression [1], cellular differentiation [2], development [3], and stress responses [4]. In plants, DNA methylation is categorized in terms of site classes as CG, CHG, or CHH (H = A, C, or T) based on the sequence context accompanied by the methylated C (mC). Methylation at mCG is maintained by methyltransferase 1, methylation at mCHG is maintained by chromomethylase 3, and methylation at mCHH is maintained by chromomethylase 2. According to enrichment tests for each site class, at least five classes of methylated genes, including unmethylated, gene body methylated, transcriptional start, CG/CHG genes, and CHH/RdDM genes, have been determined [5]. The methylated gene body is evolutionary conserved, while the transcriptional start CG/CHG and CHH/RdDM genes are not conserved. 

In-vitro culture induces somaclonal variations (SVs) in plants [6]. Several studies show that DNA methylation can lead to SVs [7,8,9,10]. DNA methylation/demethylation is affected by in-vitro propagation, which comprises a dedifferentiation (callus formation) process followed by a re-differentiation (plant regeneration) course [11]. González showed that the DNA methylation pattern may exert different effects because the in-vitro culture is treated as a stressful condition [12]. The level of DNA methylation in the SVs of oil palm is lower than that of normal regenerants, leading to mantled inflorescence syndrome and decreased production. Further studies suggest that oil palm production could be restored when methylation occurs extensively near the *Karma* splice site [13]. 

The SVs of pineapple have been identified successively since Wakasa’s finding of many variants from in-vitro cultures in Smooth Cayenne [14,15,16]. Variations in pineapple from in-vitro culture have been documented at the morphological, biochemical, chromosomal, molecular, and epigenetic levels; these variants include leaf color, spine, wax, and inflorescence characteristics [14,17]. Two new pineapple somaclones (i.e., P3R5 and Dwarf) have been derived from the in-vitro culture of ’Red Spanish’ pineapple [18], and the concentration of chlorophyll, contents of phenolics, specific activity of superoxide dismutase, and thickness of leaf photosynthetic parenchyma show differences between donor and variants. Other results reveal different Amplified Fragment Length Polymorphism(AFLP) banding patterns in both variants [18]. Moreover, using flow cytometry, fewer polyploid and aneuploid variants were identified in vitro-regenerated plants compared with donor plantlets. Also, supplementation of vitro-media with high concentrations of Benzylaminopurine (BAP), abscisic acid (ABA) and NaCl can produce dwarf plantlets, but these can recover to the normal phenotype after transfer to Murashige and Skoog (MS)media. In addition, the histone deacetylase activity of dwarf plantlets has been shown to be higher than that of non-dwarf plantlets. These observations suggest epigenetic modulation might affect SV in pineapple [19,20].

Several approaches such as methylation-sensitive amplification polymorphism, transposon methylation display, and RAD-sequencing, have been used to analyze DNA methylation [21,22]. The results obtained generally indicate that DNA methylation is an important source of SV. Although many methods for studying DNA methylation are available, whole genome bisulfite sequencing (WGBS) is considered the best tool for studying this phenomenon in plants because the technology allows single-nucleotide (*N*) resolution of 5-methylcytosine on a genome, which other techniques do not. WGBS has been used to sequence many methylated plant genomes, and the results of comparative epigenetics suggest that DNA methylation varies across species [23]. In this study, the DNA methylation profiles of cutting seedlings (CK) and five SV pineapple variants were investigated by WGBS. 

## 2. Materials and Methods

### 2.1. Plant Materials

The pineapple cultivar was ‘Tai Nong 21’. The cutting seedlings were produced from a leaf with bud of crown buds. Then the 5 suckers were used as the explants for regenerations. After surface sterilization, the explant was placed on the MS + 0.5 mg/L BAP + 2.5 mg/L naphthylacetic acid (NAA) for induction of the callus and then the regeneration plants were induced by MS + 2.0 mg/L BAP + 1.0 mg/L NAA. The regeneration plant was transplanted into the field located at the South Subtropical Crop Research Institute Chinese Academy of Tropical Agricultural Sciences (21°10′2″ N; 110°16′34″ E). The five variants were derived from callus culture. The phenotype of CK and 5 SV plants including WS (white-striped leaves), YS (yellow-striped leaves), GS (green-striped leaves), LS (spiny leaves), and TP (tetraploid seedling) are showed in Figure 1. DNA was extracted from leaves by using the CTAB method. 

### 2.2. Construction and Sequencing of DNA Libraries

Genomic DNA were fragmented with a mean size of 100–300 bp by sonication (Covaris) and purified with MinElute PCR Purification Kit (QIAGEN). The fragmented DNAs were end repaired and a single “A” nucleotide was added to the 3′ end of the blunt fragments. Then the genomic fragments were ligated to methylated sequencing adapters. Fragments with adapters were bisulfite converted using Methylation-Gold kit (ZYMO). Finally, the converted DNA fragments were PCR amplified and sequenced using Illumina HiSeqTM 2500 by Gene Denovo Biotechnology Co. (Guangzhou, China). Sequencing was conducted using Illumina HiSeq™ 2500 (Gene De Novo Biotechnology Co., Guangzhou, China) according to the manufacturer’s instructions. During the library construction, a known λ sequence which does not contain the mC was treated by the bisulfite as a spike-in control to evaluate the C to U conversion rate.

### 2.3. Data Filtering and Read Alignment

To obtain high-quality, clean reads, low-quality reads and contaminating DNA were filtered from raw reads. Reads with any of the following characteristics were removed: (1) containing an adaptor sequence, (2) *N* number > 10%, and (3) low-quality reads containing >40% low-quality (*Q*-value ≤ 20) bases. The obtained clean reads were mapped to the pineapple genome sequence V3 which is available online (http://pineapple.angiosperms.org/pineapple/htmlindex.html), by using BSMAP software [24] (version 2.90). Effective coverage of specific analysis area = 100 * (number of C bases effectively covered in the area/number of C bases in the area).

### 2.4. Analysis of Methylation Level

Methylation level was determined by dividing the number of reads covering each mC by the total number of reads covering that cytosine, which is also equal to the mC/C ratio at each reference cytosine. The methylation level was calculated based on methylated cytosine percentage in the whole genome, in each chromosome and in different regions of the genome for each sequence context (CG, CHG and CHH). Additionally, mC percentage was used to assess the methylation level of the whole genome. The methylation profile of flanking 2 kb regions and gene bodies (or transposable elements, TEs) was plotted on the strength of the average methylation level for each 100 bp interval. The methylation level for each sequence context (CG, CHG, and CHH) were analyzed in the gene body, Upstream_2k, Downstream_2k, Transposon element, Exon, Intron, CDS, Five_UTR, and Three_UTR.

### 2.5. Differentially Methylated Regions (DMR) Analysis

Differentially methylated regions for each sequence context (CG, CHG and CHH) between SV and CK were identified according to the following stringent criteria: (a) more than five methylated cytosines in at least one sample; (b) more than ten reads coverage for each cytosine, and more than four reads for each methylated cytosine; (c) region length is between 40 bp and 10 kb; (d) the distance between adjacent methylated sites < 200 bp; (e) the fold change of the average methylation level >2; (f) Pearson’s chi-square test (χ^2^) value *P* ≤ 0.05. The putative DMRs overlapping at adjacent 2kb (upstream or downstream) or body regions of genes or transposable elements (TEs) were identified for further study. The analysis first maps all DMR-related genes to the biological process, cellular component, and molecular function terms in the Gene Ontology (GO) database (http://www.geneontology.org/) by blast2go. This calculates the number of genes for each term, and then applies a hypergeometric test with corrected *p*-value ≤0.05 to find out the genotype. The GO entries were significantly enriched DMR-related genes. Pathway significance enrichment analysis using Kyoto Encyclopedia of Genes and Genomes (KEGG) pathway as a unit applies hypergeometric tests with corrected *p*-value ≤ 0.05 to find pathways that are significantly enriched in DMR-related genes compared to the entire genomic context. The figures were drawn by R language.

## 3. Results

### 3.1. Generation of Pineapple Methylation Data

After data filtering, a total of 137,934,206 clean reads for CK, 133,967,826 clean reads for WS, 220,440,122, clean reads for YS, 122,918,554 clean reads for GS, 124,401,174 clean reads for LS and 113,400,154 clean reads for TP were generated respectively. The clean reads were mapped to sequences that are unique in the genome after bisulfite conversion from every possible methylation pattern, and the mapping rates ranged from 83.31% to 86.03% (Appendix A). The total sites of C was 117,404,024, and the actual coverage rates ranged from 81.27% to 84.05% in chromosomes (Appendix A). The coverage rate of CG ranged from 79.19% to 82.71%, and the CHG ranged from 81.94% to 83.35%, while CHH ranged from 81.60% to 84.22 % (Appendix A).

### 3.2. DNA Methylation Pattern Analysis

Cytosine methylation of pineapple occurs in three sequence contexts, including CG, CHG, and CHH (H = A, T, or G). During the library construction, a known λ sequence which does not contain the mC was treated by the bisulfite as a spike-in control to evaluate the C to U conversion rate. The C to U conversion of the 5 samples was >99.2%. The total number of the mC 23,958,008, 23,931,882, 22,433,413, 23,217,734, 22,383,039, and 22,441,480 in the CK, WS, YS, GS, LS, and TP respectively. Measured mCG levels ranged from 66.31% in the YS to 73.90% in the GS. Also, the measured mCG levels ranged from 41.39% in the YS to 45.38% in the GS. The measured mCHH levels ranged from 11.06% in the LS to 13.63% CK (Figure 2A). Among the six materials studied, mCG showed the highest levels whereas mCHH showed the lowest levels of DNA. The proportions of mCG, mCHG, and mCHH were analyzed, and mCHH revealed the highest proportion among the six samples (Figure 2B). Per-site methylation levels are shown in Figure 2C. The mCG and mCHG levels of most samples reached 90−100%. By comparison, the mCHH levels of these samples were in the range of 10–40%.

In all six pineapple samples, the methylation profiles in different motifs of genes (gene body), promoters (2 kb upstream), transcriptional termination region (2 kb downstream), exon, introns, coding sequences (CDs), 5′-untranslated regions (UTRs, and 3′-UTR), suggest that the methylation level is distinguished in different genome regions. Compared with those of CK, the methylation levels of mC, mCG, mCHG, and mCHG in the five SV plants decreased in the gene body, 2 kb flanking sequences, and intron. By comparison, mC, mCG, mCHG, and mCHG in GS showed the highest methylation levels in the exon, CDs, 3-UTR, and 5-UTR. Among the six plant variations, methylation level showed the order 2 kb upstream > 2 kb downstream > intron > gene body > CDs (Figure 3).

The characteristics of the base sequence flanking mC (postion 4) were analyzed. Compared with that of CK, changes in mCG in the SV plants were mainly observed at position 1 or 9. In mCHG, the base probability of CK, YS, GS, LS, and TP was C followed by T and A at position 5, and the probability of WS was C followed by A and T. In mCHH, the base probabilities of the six samples at positions 5 and 6 were A followed by T and C and T followed by A and C, respectively (Figure 4).

### 3.3. Analysis of Differentially Methylated Regions 

Genome-wide methylation patterns were assessed between the 5 SV plants. Differentially methylated regions (DMRs) were identified to analyze context-specific DNA methylation levels in each sample relative to those of CK.

Among the groups, a total of 107,250 DMRs were identified in the plant genome. The numbers of hypomethylated DMRs in WS showed the order CG > CHG > CHH. In the 5 SV plants, the DMRs of CG were more likely to be hypermethylated and the DMRs of CHG and CHH DMRs were more likely to be hypomethylated (Figure 5). The distributions of methylation status in the genomic regions demonstrate that approximately 9411 hypermethylated and 4518 hypomethylated DMRs exist between CK and WS. Indeed, the number of hypermethylated DMRs was approximately twice that of hypomethylated DMRs between CK vs. WS, CK vs. YS CK vs. GS and CK vs. LS. The distribution of DMRs reveals that hypermethylated DMRs are mostly found in the CG motif in the gene body. In comparison with hypermethylated DMRs, hypomethylated DMRs were mostly found in the CG and CHG motifs and distributed in the gene body (Figure 6).

DMRs of 5 SV and 1 CK plant were assessed. DMRs were considered consistent if more than two groups exhibited, while exhibiting in only one sample was considered rare DMRs. The results showed that total numbers of rare DMRs were 51,682 in hypermethylated, while approximately 5614 DMRs were consistent. In hypermethylation, rare DMRs were more than 9 times that of consistent DMRs, and total numbers of rare DMRs were 71,131 in hypomethylated, while approximately 3264 DMRs were consistent. In hypomethylation, rare DMRs were more than 20 times that of consistent DMRs (Table 1).

There are 186, 31 and 84 DMRs in the mCG, mCHG and mCHH context respectively which was conserved in the 5 SV of pineapples (Figure 7A). The heatmap result showed that most of level of methylation among the 186 conserved mCG DMRs increased in the 5 SV plant. However, the level of methylation was decreased in the CK among the 186 conserved mCG DMRs. In the mCHG and mCHH context, most of the level of methylation decreased in the SV plants and increased in the CK (Figure 7B).

### 3.4. Enrichment Analysis of DMR-Related Genes

The DMR genes were annotated to GO and KEGG databases to analyze their functions. GO analysis revealed that the same biological processes, cellular components, and molecular functions were annotated in CG, CHG, and CHH (Appendix A). In terms of biological process, DMGs were mainly enriched in cellular process, metabolic process, and single-organism process. In terms of cellular component, DMGs were mainly enriched in cell, cell part, and organelle. In terms of molecular function, DMGs were mainly enriched in binding and catalytic activity. KEGG analysis revealed that DMGs for the CG context were significantly enriched in biosynthesis of secondary metabolites, starch and sucrose metabolism, pentose and glucuronate interconversion (Appendix A). We also found that some DMGs are involved in biological processes that are important in response to stress, including abiotic stimulus (GO: 0009582), response to hormone (GO: 0009725), response to stress (GO: 0006950), response to inorganic substance (GO: 0010035), and DNA modification (GO: 0006304). This finding suggests that DNA methylation of these specific genes could affect the biosynthesis of secondary metabolites and stress functions (Appendix A). Detailed results of the GO and KEGG analyses of CG, CHG, and CHH are provided in Appendix A

### 3.5. The Conserved DMR-Related Genes between the 5SV Plants

There are 824, 675 and 0 DMR-related genes in the mCG, mCHG and mCHH context respectively between the 5 SV of pineapples (Figure 8A). The KEGG result showed that the DMR-related genes in the mCG context was most enrichment in the metabolic pathway and second metabolic pathway. In the mCHG context the DMR-related genes were most enrichment in the spliceosome and plant–pathogen interaction pathway (Figure 8B). 

## 4. Discussion

SVs include morphological, cytological, biochemical, and genetic/epigenetic changes induced by tissue culture [25]; alterations in cytosine methylation were detected in all cases studied, thus suggesting that DNA methylation may play an important role in the mutation process [4,26,27,28,29]. Scherer et al. found that DNA methylation level is associated with the density of newly formed shoot apical meristems [19]. The change of the DNA methylation of the SV plants which regenerate to form the callus culture remains unclear. In this study, we used BS-sequencing to investigate the DNA methylation profiles of the pineapple genome to analyze the methylation level of the SV plants. The changes of DNA methylation have been reported in the regenerated when compared to the values obtain from the seed derived plants or cutting seedling. In the *Arabidopsis* the regenerated plant has significantly higher average methylation values when compared with the seed derived plants [21]. 56.22% of cytosines in the CHH context, 27.06% in the CG context and 16.67% in the CHG context were methylated in pineapple (Figure 2B). Hence, methylation patterns vary in plants. Changes in DNA methylation are a main cause of SV in plants. Changes in DNA methylation have been observed in abnormal sugarcane plantlets. The rate of hypomethylation is higher than that of hypermethylation [30].

In the present study, the total methylcytosine level was 27.98% in CK which is higher than that in SV plants. Compared with CK, most proportions of CG and CHG increased but proportions of CHH decreased in SV plants. The results suggest that the methylation patterns are not changed; instead, the proportion of the three contexts appears to vary. This finding indicates that changes in the three contexts may cause variations in methylation. Stroud et al. [31] showed that the hypomethylation in the regenerated rice was higher than that of CK and led to the regenerated rice had significant loss methylation compared to the CK. During the callus culture steps, the methylation level was wild type > callus > regenerated plant. The result indicated that the loss of methylation was induced in the dedifferentiation and redifferentiation. Hsu et al. [32] showed that there were dynamics of the methylome during the rice regeneration program. The changed of methylation levels at a high proportion, which occur consistently in multiple independent callus culture lines of maize. Hence, epigenetic variation could favor some regions or loci [33]. Similar results wherein methylation changes were shared at the same locus have been observed in regenerated rye plants [12]. DNA methylation is consistently widespread in the CG or CHG contexts [34]. In the present study, we analyzed DMRs between CK and the plant variants and found that changes in DNA methylation are present in CG and CHG at high and low levels, respectively. As shown in Figure 3, methylation mainly occurs in the gene body. The high rate of gene body changes to DNA methylation levels may directly affect the gene expression and lead to the phenotypic variation in the regenerated pineapple derived from callus culture. In a further study, the gene expression will be analyzed by RNA-seq approach.

Variations in DNA methylation, which is a non-random mutation, are induced by callus culture [25]. The methylation sites in regenerated plants are non-random behaviors in rye genomic methylation [12]. Han et al. observed 303 DMRs of hypomethylated common to more than six or more samples, which was less than random chance [33]. Although most of the methylation events were random, several sites appeared particularly susceptible [31,32]. The methylation effects on the SV in those conserved sites is not known. In the present study, the total number of the hypomethylated DMRs was 74,594 and hypermethylation DMRs was 57,296 in the 5 SVs. The result in consistent with the result that the total methylation level of mC decreased in the 5 SV plant. Moreover, there are 5614 consistent DMRs was hypermethylated (10.86%) and only 3264 consistent DMRs was hypomethylation (4.59%). The result suggests that the consistent DMRs methylation level was increased. Also, the heatmap showed that the level of methylation of conserved DMR in the mCG context increased when compared with the CK. The rate of consistent/rare was 0.075. This indicates that methylation at that site was random. A similar result was obtained in the rice [31,32].

## 5. Conclusions

The methylation level was various between the CK and 5 SV plants and the GO of the methylated gene result showed that callus culture of pineapple induced DNA methylation may result in the SV.

## Figures and Tables

**Figure 1 genes-10-00877-f001:**
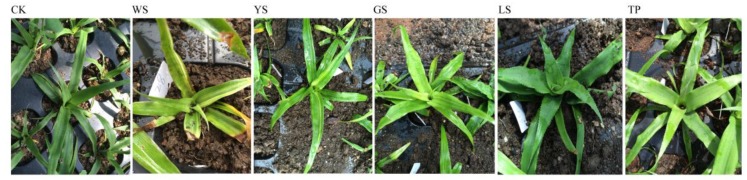
The phenotype of the CK and somaclonal variation (SV) plants. CK was the cutting seedings. WS (white-striped leaves), YS (yellow-striped leaves), GS (green-striped leaves), LS (spiny leaves), and TP (tetraploid seedling) are the SV plants of pineapple.

**Figure 2 genes-10-00877-f002:**
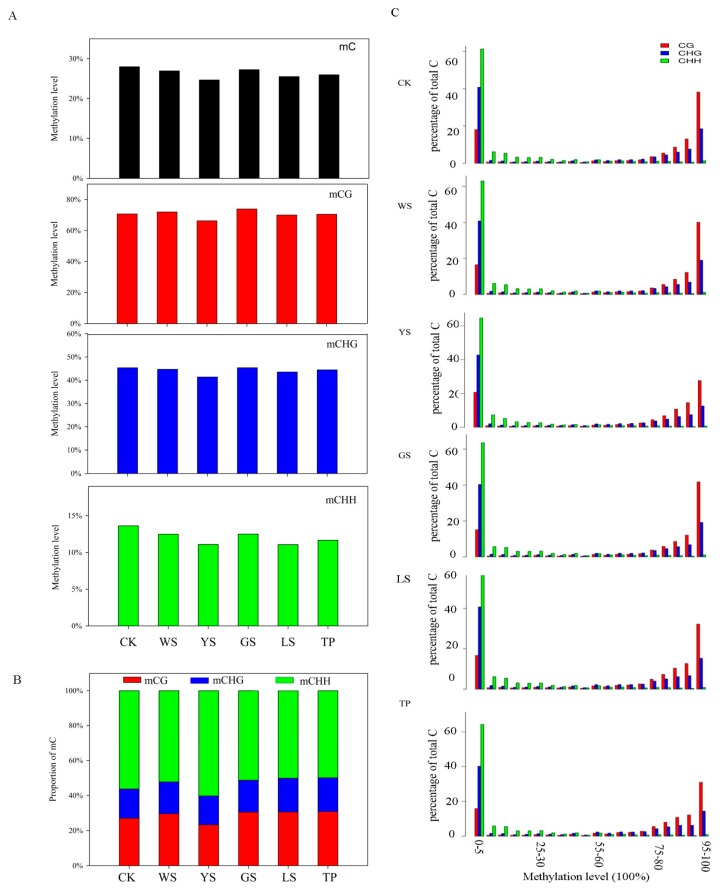
DNA methylation pattern in SV plants. (**A**) Methylation levels of SV plants. According to the characteristics of sequence, the mC are classified into CG, CHG, and CHH (H = A/C/T). (**B**) Relative proportions of mCs in the sequence contexts of CG, CHG, and CHH. (**C**) Distribution of methylation levels of mCs in each sequence context. Only mCs covered by at least five reads were used to calculate the methylation levels. The *x*-axis represents the percentage of reads showing mCs at a reference cytosine site, while the *y*-axis indicates the fraction of total mCs calculated within bins of 10%.

**Figure 3 genes-10-00877-f003:**
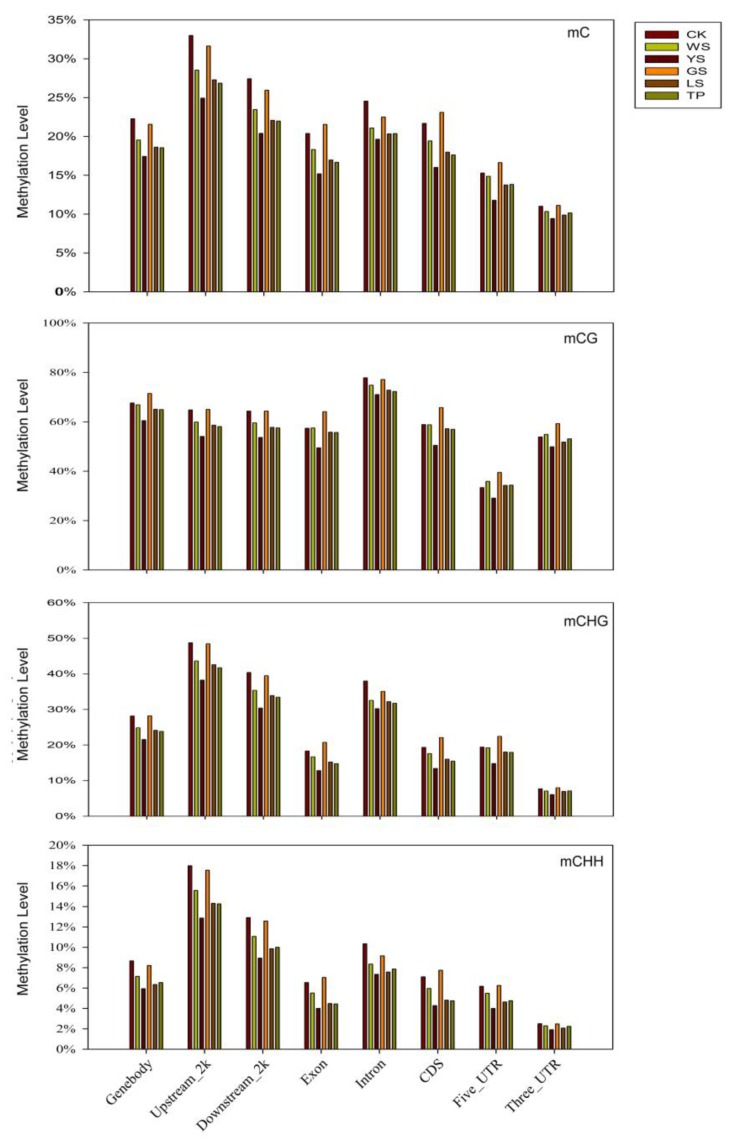
Relative methylation level in three contexts for different genomic regions in SV plants. The *x*-axis shows the sequence contexts, while the *y*-axis shows the methylation level.

**Figure 4 genes-10-00877-f004:**
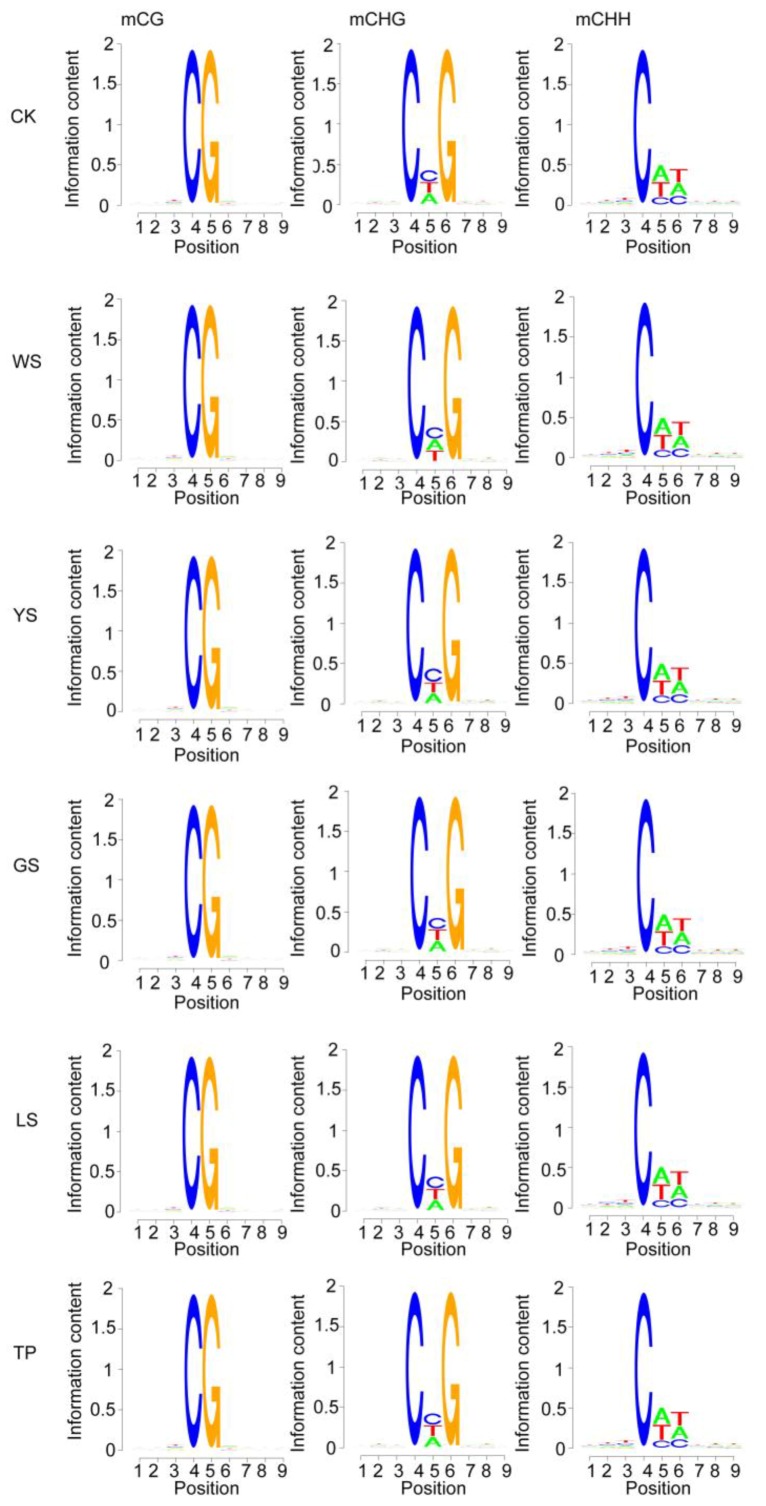
Methylation preferences in the 9 bp region spanning mCG, mCHG, and mCHH sites in SV plants. H = A/C/T. The *x*-axis shows the base number of the methylation site.

**Figure 5 genes-10-00877-f005:**
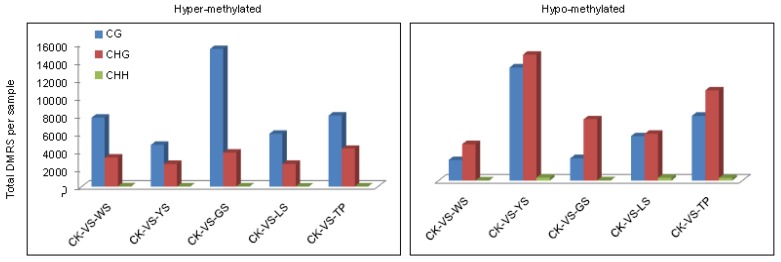
Number of differentially methylated regions (DMRs) per context for each combination in CK and SV plants. Bars indicate the total number of hypermethylated and hypomethylated DMRs for each variant compared with those of the controls.

**Figure 6 genes-10-00877-f006:**
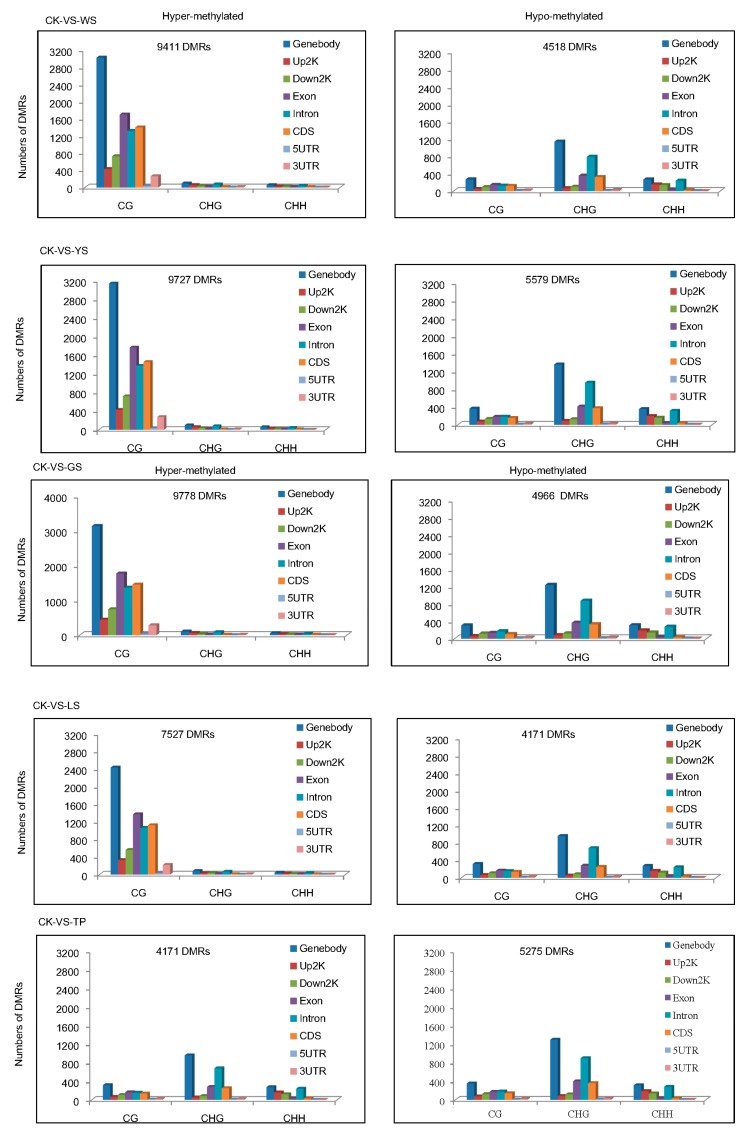
Numbers of DMRs per context for each combination in CK and the SV variants. Bars indicate the total number of DMRs for both hypermethylated and hypomethylated DNA compared with those of the controls.

**Figure 7 genes-10-00877-f007:**
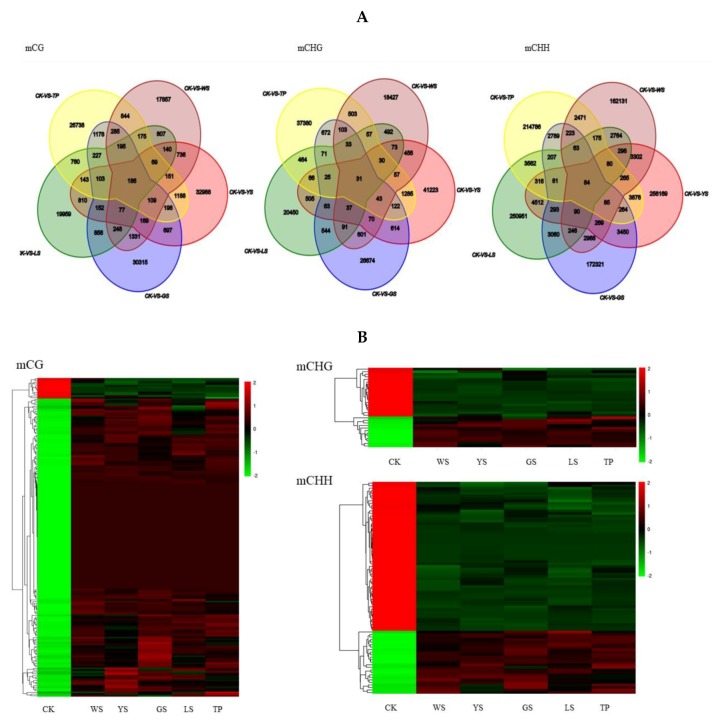
The conserved DMR between the 5 SV plants of pineapple. (**A**) Venn diagram showing the numbers of DMR between the 5 SV plants of pineapple. (**B**) The heatmap of level of methylation of conserved DMR between the 5 SV plants of pineapple.

**Figure 8 genes-10-00877-f008:**
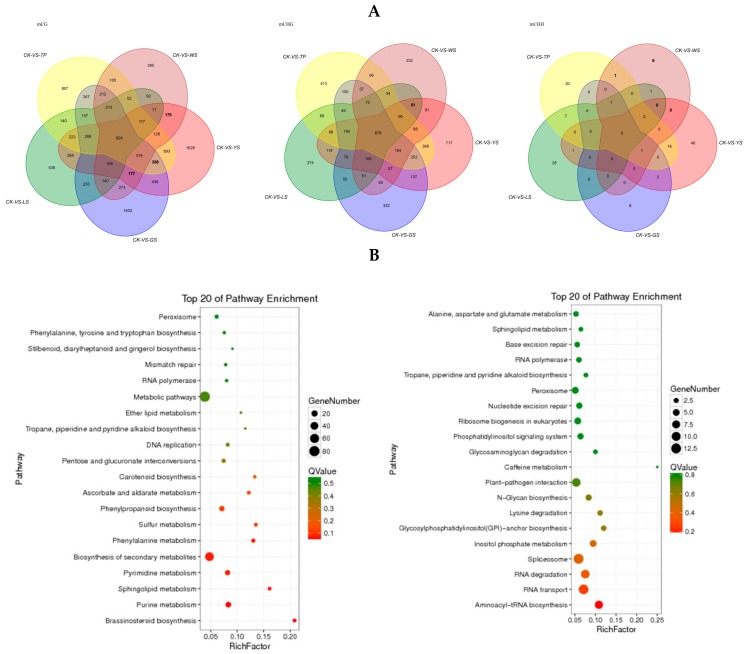
DMR-related genes in the gene body that overlapped between the 5 SV plants of pineapple. (**A**) Venn diagram showing the numbers of DMR-related genes in the gene body; (**B**) the KEGG pathway of the DMR-related genes in the gene body.

**Table 1 genes-10-00877-t001:** Numbers of DMRs in the CG, CHG, and CHH contexts.

Context of mC	Rare DMRs		Consistent DMRs	
	Hypermethylated	Hypomethylated	Hypermethylated	Hypomethylated
CG	36,475	29,657	4534	1366
CHG	15,085	40,480	1078	1890
CHH	122	994	2	8
Total numbers	51,682	71,131	5614	3264

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
