# Peer review of "Whole-Genome Bisulfite Sequencing Reveals a Role for DNA Methylation in Variants from Callus Culture of Pineapple (*Ananas comosus* L.)"

_genes, 2019, doi:10.3390/genes10110877_

Round 1
Reviewer 1 Report
The paper provides a comparison of DNA methylation profiles of 5 somaclonal variations (SV) of pineapple among each other and with leaves collected from cutting seedlings.
Changes to the DNA methylation pattern are one of the classic examples of such variations. Whole-genome bisulfite sequencing has already been used to describe epigenome changes which occur as a result of regeneration from tissue culture (Stroud et al., Elife, 2013; Hsu et l., Epigenomes, 2018). The authors don't cite these papers while having a comparison of the results on pineapple and on other plants might result in a better understanding of the phenomenon.
Although the paper provides valuable insights into the effect of regeneration on the DNA methylation variations, it has several flaws.
Each sample had only one replica. The authors didn’t use any spike-in controls or another method to validate the C to U conversion rate and this can affect the results. Although the method is called “Whole-genome” bisulfite sequencing, it doesn’t necessarily provide coverage for every single C in the genome. While authors provide redundant and excessive data (the number of the clean reads is shown twice, “Clean data” is a useless parameter for the paper) on the number of reads, it is unclear what is the actual fraction of genome covered in the analysis and how does the distribution of the coverage per site looks like. Only sites covered in all samples should be considered for analysis. Which tool or method was used to measure the enrichment? With which parameters? Multiple conclusions are made without applying any statistical test. Please, perform an appropriate statistical test for every comparison you are making. Despite the claim in the abstract saying “This study provided insights into the potential mechanisms of SVs in pineapple” there is nothing like this in the actual paper. Figures have low resolution and sometimes small font. In some cases, the figure labeling doesn’t look like a proper one. For example, it should be “Number of genes” instead of “Num of Genes”. The Fig. S1 legend says: “Gene numbers and percentages (on the log scale) are listed for each category”, but it doesn’t seem to fit the actual figure. The raw and processed data hasn’t been uploaded to any repository making it impossible to verify the claims made in the manuscript. The main question of the paper is how the CV variants differ from each other, but there is no comparison to what happens between different cutting seedlings. Are these regions/genes more prone to changes overall or is it something specific to CV? The manuscript is poorly written. It contains multiple typos, errors, and sentences which seem to make no sense. For example: "A total of 27.98% of CK genomic cytosines were methylcytosines which was more than 5 SV plants. The mCG and mCHG were hyper-methylation, mCHH were hypo-methylation", "And mehtylation at mCHH was maintained by chromomethylase2", "Moreover, compared with donor plants, there are fewer polyploid or aneuploid variants were evaluated by flow cytometry", "DNA methylation variation is induced by tissue culture, which is a non-random mutation" etc.Author Response
Please see the attachment

Reviewer 2 Report
The study requires incorporation of statistical analysis to make comparisons between SV's.
Also, it is expected there might be other issues that could be explored with this large data-set. As an example, the SV's associated with chloroplasts and leaf spines are common variants arising from tissue culture. These are not random but highly predictable. Perhaps the authors can explore the nature of this phenomena.
SV is a striking feature of pineapple and, combined with clonal reproduction, has contributed much to the heterogeniety of the genome and the crop's domestication. SV in pineapple fields is a major source of clonal deterioration and costs associated with maintaining purity of clonal lines is high. An improved understanding of SV is keenly sought.
Specific Comments
Line 30. Comma out of place.
Line 32. ‘mehtylation’ should be spelt ‘methylation’
Line 34 needs a space between ‘gene’ and ‘body’
Line 43 Should read ‘DNA methylation can lead to SV.’
Line 44. Should read ‘is restored when the methylation becomes dense near the Karma splice site.’
Line 50 What is ‘Red Spanish Pinar’. Should it just be, ‘ ‘Red Spanish’ pineapple.’ Usually varietal names are in apostrophes unless they are preceded by the term cultivar. So, either, The pineapple cultivar, Red Spanish. Or ‘Red Spanish’ pineapple.
Line 54. Space needed after comma. Also, the sentences starting on line 53 do not make sense. Should it be, ‘Moreover, using flow cytometry, fewer polyploid and aneuploid variants were identified in vitro-regenerated plants compared with donor plantlets. Also, supplementation of vitro-media with high concentrations of BAP, ABA and NaCl can produce dwarf plantlets, but these can recover to the normal phenotype after transfer to MS media. In addition, the histone deacetylase activity of dwarf plantlets has been shown to be higher than that of non-dwarf plantlets. These observations suggest epigenetic modulation might affect SV in pineapple. ’
Line 59. ‘Several approaches..’
Line 61. ‘Those results indicate..’
Line 63. ‘..considered as the best tool for studying..’
Line 70. More description is need for the plant materials. What are ‘cutting seedlings’? Are they clonal plants arising from individual seedlings such as slips or suckers? Or were they produced from individual buds on the stem by dissecting the stem? How many? Are the variants all variants from one line or are there several seedling clones each with a different variant?
Is the normal phenotype spiny or spiny-tip or another leaf margin morphology?
Line 86. Please include the genome version as this is subject to change soon.
Line 116. I read this to say CK plants have the highest number of methylated sites. Is this correct?
Lines 114-122 There are no statistics mentioned here comparing the various treatments. It is not possible to conclude differences if there are no statistical comparisons that estimate variance. This needs a biometrician to examine the data and see if some simple analyses can be done. Perhaps data for the different variants could be treated as replications?
The writing style is mostly easily understood but does requires some revision to improve the English.
Round 2
Reviewer 1 Report
The manuscript has been significantly improved.
However, some issues remain:
1) There is no statistical analysis. The authors report that some DMRs were detected in multiple plants, but is this significant?
2) Which tool and which criteria were used for the functional enrichment analysis?
3) The analysis complementary to the DMR detection seems missing. What are the regions which are the most conservative between the plants, which genes are enriched in those? Are these genes more evolutionary conserved?
4) Some of the figures still have resolution/font/formatting problems:
Fig S2 – low resolution, hard to read Fig 3, Fig 5, Fig 6 Figure defects5) Multiple phrases are hard to understand and/or contain typos:
Table S1: Should be “sequencing depth”
270 online
271 .
373 sites
Multiple lines: spike-in
271 It is unclear if “coveraged” a new term introduced by the authors or just a typo and it should be “covered”.
Effective coverage of a specific area is introduced but never mentioned again in the manuscript.
The definition of “Methylation level” in line 278 is very confusing. Effective sequencing reads are mentioned without specifying what it means. It seems like it is the same as “clean” described in 2.3. Bisulfite sequencing doesn’t allow to directly see methylcytosines, so it would be better to write that it is the number of cytosines which is included in the analysis and to mention that methylated cytosines are not affected by bisulfite treatment.
296-297 hard to read and follow. It would be better to have numbers next to the plants.
639 Should be “DMRs of 3 SV plants were compared to CK”
685 It is unclear if the methylation levels present here correspond to Arabidopsis or to pineapple. Is this a comparison of different context methylation within one plant or between the plants? This part should be expanded.
801 “The high rate of gene body changes to DNA methylation levels for epigenetic variations were probably directly relevant to the gene expression variation that manifested as phenotypic variation in pineapple derived from tissue culture.” - Could you please reformulate this statement? It seems too strong for the presented evidence.
805 “The methylation changes in regenerated plants are non-random behaviors in rye genomic methylation" - This sentence is hard to understand. At the same time the paper cited here is very relevant to this manuscript. How do rye DMRs correspond to pineapple DMRs? Does it happen to the same genes?
809 “but several of site was susceptibly to be methylation” – typos
810 “In the present study, approximately 5,614 and 3,264 810 consistent DMRs, accounting for 10.86% and 4.59% of total DMRs, respectively were present in hyper/hypomethylation.” - It is unclear what the authors mean here.
Author Response
There is no statistical analysis. The authors report that some DMRs were detected in multiple plants, but is this significant?Answer:There is no replication in the present study, so we don’t how to do the statistical analysis. But we analysis the conserved DMRs in the 5 SV of pineapples. The result showed as follow.
There are 186, 31 and 84 DMRs in the mCG, mCHG and mCHH context respectively which was conserved in the 5 SV of pineapples. (Fig 7A). The heatmap result showed that most of level of methylation among the 186 conserved mCG DMRs increased in the 5 SV plant. But the level of methylation was decreased in the CK among the 186 conserved mCG DMRs. In the mCHG and mCHH context the most of the level of methylation decreased in the SV plants and increased in the CK (Fig 7B).
A
B
Fig 7 The conserved DMR between the 5 SV plants of pineapple. A, Venn diagram showing the numbers of DMR between the 5 SV plants of pineapple. B, The heatmap of level of methylation of conserved DMR between the 5 SV plants of pineapple
Which tool and which criteria were used for the functional enrichment analysis?Answer: The analysis first maps all DMR-related genes to the biological process, cellular component, and molecular function terms in the Gene Ontology database (http://www.geneontology.org/) by blast2go . This calculates the number of genes for each term, and then applies a hypergeometric test with corrected p-value ≤ 0.05 to find out the genotype. The GO entries were significantly enriched DMR-related genes . Pathway significance enrichment analysis using KEGG pathway as a unit applies hypergeometric tests with corrected p-value ≤ 0.05 to find pathways that are significantly enriched in DMR-related genes compared to the entire genomic context .The figure was draw by R language
3) The analysis complementary to the DMR detection seems missing. What are the regions which are the most conservative between the plants, which genes are enriched in those? Are these genes more evolutionary conserved?
Answer: There are 824, 675 and 0 DMR-related genes in the mCG, mCHG and mCHH context respectively between the 5 SV of pineapples. (Fig 8A). The KEGG result showed that the DMR-related genes in the mCG context was most enrichment in the metabolic pathway and second metabolic pathway. In the mCHG context the he DMR-related genes was most enrichment in the spliceosome and plant-pathogen interaction pathway.
A
B
Fig 8 DMR-related genes in the genebody that overlapped between the 5 SV plants of pineapple. A Venn diagram showing the numbers of DMR-related genes in the genebody; B the KEGG pathway of the DMR-related genes in the genebody
4) Some of the figures still have resolution/font/formatting problems:
Fig S2 – low resolution, hard to read Fig 3, Fig 5, Fig 6 Figure defects
Answer: The Fig S2 was revised. But we don’t know how to revised the Fig 3, Fig 5, Fig 6.
5) Multiple phrases are hard to understand and/or contain typos:
Table S1: Should be “sequencing depth”
270 online
271 .
373 sites
Multiple lines: spike-in
271 It is unclear if “coveraged” a new term introduced by the authors or just a typo and it should be “covered”.
Answer : Thanks!
Effective coverage of a specific area is introduced but never mentioned again in the manuscript.
Answer : the effective coverage of the specific analysis are =100*(number of C bases effectively covered in the area/ number of C base in the area).The analysis of mCG, mCHG and mCHH was based on Effective coverage.
The definition of “Methylation level” in line 278 is very confusing. Effective sequencing reads are mentioned without specifying what it means. It seems like it is the same as “clean” described in 2.3. Bisulfite sequencing doesn’t allow to directly seeing methylcytosines, so it would be better to write that it is the number of cytosines which is included in the analysis and to mention that methylated cytosines are not affected by bisulfite treatment.
Answer : The methylation level was calculated based on methylated cytosine percentage in the whole genome, in each chromosome and in different regions of the genome for each sequence context (CG, CHG and CHH).Thanks.
296-297 hard to read and follow. It would be better to have numbers next to the plants.
Answer : After data filtering, a total of 137,934,206 clean reads for CK, 133,967,826 clean reads for WS, 220,440,122, clean reads for YS, 122,918,554 clean reads for GS, 124,401,174 clean reads for LS and 113,400,154 clean reads for TP were generated respectively.
639 Should be “DMRs of 3 SV plants were compared to CK”
Answer : Thanks
685 It is unclear if the methylation levels present here correspond to Arabidopsis or to pineapple. Is this a comparison of different context methylation within one plant or between the plants? This part should be expanded.
Answer : The changes of DNA methylation have been reported in the regenerated when compared to the values obtain from the seed derived plants or cutting seedling. In the Arabidopsis the regenerated plant has significantly higher average methylation values when compared with the seed derived plants[21]. Thanks
801 “The high rate of gene body changes to DNA methylation levels for epigenetic variations were probably directly relevant to the gene expression variation that manifested as phenotypic variation in pineapple derived from tissue culture.” - Could you please reformulate this statement? It seems too strong for the presented evidence.
Answer : The high rate of gene body changes to DNA methylation levels may directly affect the gene expression and lead to the phenotypic variation in the regenerated pineapple derived from tissue culture. In the further study, the gene expression will be analyzed by RNA-seq approach.
805 “The methylation changes in regenerated plants are non-random behaviors in rye genomic methylation" - This sentence is hard to understand. At the same time the paper cited here is very relevant to this manuscript. How do rye DMRs correspond to pineapple DMRs? Does it happen to the same genes?
Answer : It should be the methylation sites in regenerated plants are non-random behaviors in rye genomic methylation. The paper used the MSAP markers to analysis the methylation changes in regenerated plants, So it didn’t identified the DMR. Thanks.
809 “but several of site was susceptibly to be methylation” – typos
Answer : but several of sites were susceptible methylation
810 “In the present study, approximately 5,614 and 3,264 consistent DMRs, accounting for 10.86% and 4.59% of total DMRs, respectively were present in hyper/hypomethylation.” - It is unclear what the authors mean here.
Answer : In the present study, The total number of the hypomethylated DMRs were 74,594 and hypermethylation DMRs was 57,296 in the 5 SVs . The result in consistent with the result that the total methylation level of mC decreased in the 5 SV plant. Moreover, there are 5,614 consistent DMRs was hypermethylated(10.86%) and only 3,264 consistent DMRs was hypomethylation(4.59%). The result suggests that the consistent DMRs methylation level was increased. Also the heatmap showed that the level of methylation of conserved DMR in the mCG context increased when compared with the CK. The rate of consistant/rare was 0.075.This indicate that the methylated site was random. The similar result obtained in the rice[31-32].

Reviewer 2 Report
The issue of a statistically sound experimental design limits what can concluded from the data. The variants obtained from in-vitro culture are essentially representatives from a single treatment and it should be appropriate to use these as replications to provide descriptive statistics for plants with a mutation. This would allow some discussion of the extent of methylation, type of methylation etc without making comparisons. A biometrician should be consulted on this point. The authors cannot however imply differences between the variants irrespective of funding. The data shown does not suggest differences with small variation at best. There is however a large body of data presented and this can be used along with descriptive statistics to provide a useful paper if that is acceptable to the journal.Please consider making further revisions to improve this aspect.
There are still typographical errors and poorly constructed sentances but the paper is very much improved.
The term 'tissue culture' should not be used as this is usually reserved for in-vitro culture of adventitous buds. the more correct term would be 'callus culture'.
Author Response
The issue of a statistically sound experimental design limits what can concluded from the data. The variants obtained from in-vitro culture are essentially representatives from a single treatment and it should be appropriate to use these as replications to provide descriptive statistics for plants with a mutation. This would allow some discussion of the extent of methylation, type of methylation etc without making comparisons. A biometrician should be consulted on this point. The authors cannot however imply differences between the variants irrespective of funding. The data shown does not suggest differences with small variation at best. There is however a large body of data presented and this can be used along with descriptive statistics to provide a useful paper if that is acceptable to the journal.Please consider making further revisions to improve this aspect.
Answer :There are 186, 31 and 84 DMRs in the mCG, mCHG and mCHH context respectively which was conserved in the 5 SV of pineapples. (Fig 7A). The heatmap result showed that most of level of methylation among the 186 conserved mCG DMRs increased in the 5 SV plant. But the level of methylation was decreased in the CK among the 186 conserved mCG DMRs. In the mCHG and mCHH context the most of the level of methylation decreased in the SV plants and increased in the CK(Fig 7B).
A
B
Fig 7 The conserved DMR between the 5 SV plants of pineapple. A, Venn diagram showing the numbers of DMR between the 5 SV plants of pineapple. B, The heatmap of level of methylation of conserved DMR between the 5 SV plants of pineapple
3.5 The conserved DMR-related genes between the 5SV plants
There are 824, 675 and 0 DMR-related genes in the mCG, mCHG and mCHH context respectively between the 5 SV of pineapples. (Fig 8A). The KEGG result showed that the DMR-related genes in the mCG context was most enrichment in the metabolic pathway and second metabolic pathway. In the mCHG context the he DMR-related genes was most enrichment in the spliceosome and plant-pathogen interaction pathway(Fig 8B).
A
B
Fig 8 DMR-related genes in the genebody that overlapped between the 5 SV plants of pineapple. A Venn diagram showing the numbers of DMR-related genes in the genebody; B the KEGG pathway of the DMR-related genes in the genebody
There are still typographical errors and poorly constructed sentances but the paper is very much improved.
Answer :Thanks. We revised the paper again.
The term 'tissue culture' should not be used as this is usually reserved for in-vitro culture of adventitous buds. the more correct term would be 'callus culture'.
Answer :Thanks

Round 3
Reviewer 2 Report
Brief Summary
The paper aimed to describe methylation patterns in five pineapple genotypes, each exhibiting a different mutation arising from tissue culture. It appears these were compared with clonal plants of the original seedling although the terminology used did not make this very clear. The study used whole-genome bisulfite sequencing to identify methylation patterns. Sequences associated with differentially-expressed methylated regions were annotated and broadly categorized as to likely function. Conclusion were drawn on the nature of the methylation.
Broad Comments
The study uses appropriate technology and generates a substantial data set but does not explore the data in much depth and does not incorporate any statistical analyses. Consequently, there is little reliable foundation to conclusions. The authors need to incorporate some form of statistical analyses when comparing treatments otherwise just provide descriptive statistics and do not compare treatments.
The writing style is mostly easily understood but does requires some revision to improve the English.
Specific comments
Line 38, 42, 55, 58. Suggest change ‘callus culture’ to ‘In-vitro culture’. It is appropriate to use the term ‘callus culture’ with respect to your experimental method. Other papers cited did not necessarily use callus so are more correctly described as in-vitro culture. Please check the document to ensure the correct terms are used to suit their context.
Line 86 Needs edit
Line 104 What is a methylation sit?
Line 22 Should it be ‘level instead of ‘levels’?
Line 226, 231, 335, 336 Needs edit.
Line 133 ‘sorted out’? Do you mean ‘identified’?
Line 142 Edit
Lines 179-190 Th comparative descriptions should be removed as it implies they have been compared and found to be different. The experimental procedure only allows descriptions. As an example, ‘The highest mCHG level was found in GS and the lowest mCG level was found in YS.’, could be rewritten as ‘ Measured mCHG levels ranged from xxx in YS to xxx in GS.’
Line 255 DMRs of 5 SV and 1 CK plant were assessed.
Line 305. ‘that are important in response to stress’
Line 342, 343 ’56.22% of cytosines in the CHH context, 27.06% in the CG context and 16.67% in the CHG context were methylated in pineapple (Fig 1B)’
Line 363 ‘The high rate’
Line 372. ‘In a further study’
Line 375. Edit
Line 380. ‘Although most of the methylation events were random, several sites appeared particularly susceptible.’
Line 381. ‘The methylation effects on the SV in those conserved sites is not known.’
Line 388-389. ‘This indicates that methylation at that site was random’
Line 419 Incomplete
Author Response
Dear reviewer and Ms. Zanna Zhang
I’m appreciate for review the manuscript and give us so many significative suggestions. We have revised the manuscript again. Thants
Your sincerely
Lin Wenqiu
Broad Comments
The study uses appropriate technology and generates a substantial data set but does not explore the data in much depth and does not incorporate any statistical analyses. Consequently, there is little reliable foundation to conclusions. The authors need to incorporate some form of statistical analyses when comparing treatments otherwise just provide descriptive statistics and do not compare treatments.
The writing style is mostly easily understood but does requires some revision to improve the English.
Answer: Thanks. We revised the manuscript according the reviewers comment. Thanks .
Specific comments
Line 38, 42, 55, 58. Suggest change ‘callus culture’ to ‘In-vitro culture’. It is appropriate to use the term ‘callus culture’ with respect to your experimental method. Other papers cited did not necessarily use callus so are more correctly described as in-vitro culture. Please check the document to ensure the correct terms are used to suit their context.
Answer: Thanks.
Line 86 Needs edit
Answer:The five variants derived from callus culture. The phenotype of CK and 5 SV plants including WS (white-striped leaves), YS (yellow-striped leaves), GS (green-striped leaves), LS ( spiny leaves), and TP (tetraploid seedling) were showed as Fig 1
Line 104 What is a methylation sit?
Answer:It should be the mC. We have revised it .
Line 22 Should it be ‘level instead of ‘levels’?
Answer:thanks
Line 226, 231, 335, 336 Needs edit.
Answer:
226 Genome-wide methylation patterns were assessed between the 5 SV plants
231 In the 5 SV plants, the DMRs of CG were more likely to be hypermethylated and the DMRs of CHG and CHH DMRs were more likely to be hypomethylated (Fig. 5).
335-336 The change of the DNA methylation of the SV plants which regenerate form the callus culture remains unclear. In this study, we used BS-sequencing to investigate the DNA methylation profiles of the pineapple genome to analysis the methylation level of the SV plants.
Line 133 ‘sorted out’? Do you mean ‘identified’?
Answer: Yes, Thanks
Line 142 Edit
Answer:The figures were drawn by R language
Lines 179-190 Th comparative descriptions should be removed as it implies they have been compared and found to be different. The experimental procedure only allows descriptions. As an example, ‘The highest mCHG level was found in GS and the lowest mCG level was found in YS.’, could be rewritten as ‘ Measured mCHG levels ranged from xxx in YS to xxx in GS.’
Answer:Thanks. Measured mCG levels ranged from 66.31% in the YS to 73.90% in the GS. Also the measured mCG levels ranged from 41.39% in the YS to 45.38% in the GS. . The measured mCHH levels ranged from 11.06% in the LS to 13.63% CK.
Line 255 DMRs of 5 SV and 1 CK plant were assessed.
Answer:thanks
Line 305. ‘that are important in response to stress’
Answer:thanks
Line 342, 343 ’56.22% of cytosines in the CHH context, 27.06% in the CG context and 16.67% in the CHG context were methylated in pineapple (Fig 1B)’
Answer:thanks
Line 363 ‘The high rate’
Answer:thanks
Line 372. ‘In a further study’
Answer:thanks
Line 375. Edit
Answer:We deleted the sentence.
Line 380. ‘Although most of the methylation events were random, several sites appeared particularly susceptible.’
Answer:thanks
Line 381. ‘The methylation effects on the SV in those conserved sites is not known.’
Answer:thanks
Line 388-389. ‘This indicates that methylation at that site was random’
Answer:thanks
Line 419 Incomplete
Answer: Bardini, M.; Labra, M.; Winfield, M.; Sala, F. Antibiotic-induced DNA methylation changes in calluses of Arabidopsis thaliana. Plant Cell; Tissue and Organ Culture 2003,72(2):157-162. DOI:10.1023/a:1022208302819.
